# The folklore of the "Swift" effect – lessons for medical research and clinical practice

**James M. Smoliga** *, **Kathryn E. Sawyer**

Tufts University School of Medicine, Department of Rehabilitation Sciences, Boston, Massachusetts

* james.smoliga@tufts.edu

## Abstract

Taylor Swift's presence at National Football League (NFL) games was reported to have a causal effect on the performance of Travis Kelce and the Kansas City Chiefs. Critical examination of the supposed "Swift effect" provides some surprising lessons relevant to the scientific community. Here, we present a formal analysis to determine whether the media narrative that Swift's presence at NFL games had any impact on player or team performance – and draw parallels to scientific journalism and clinical research. We performed a quasi-experimental study, using covariate matching. Linear mixed effects models were used to determine how Swift's presence or absence in Swift-era games influence Kelce's performance, relative to historical data. Additionally, a binary logistic regression model was developed to determine if Swift's presence influenced the Chief's game outcomes, relative to historical averages. Across multiple matching approaches, analyses demonstrated that Kelce's yardage did not significantly differ when Taylor Swift was in attendance (n = 13 games) relative to matched pre-Swift games. Although a decline in Kelce's performance was observed in games without Swift (n = 6 games), the statistical significance of this finding varied by the matching algorithm used, indicating inconsistency in the effect. Similarly, Swift's attendance did not result in a significant increase in the Chiefs' likelihood of winning. Together, these findings suggest that the purported "Swift effect" is not supported by robust evidence. The weak statistical evidence that spawned the concept of the "Swift effect" is rooted in a constellation of fallacies common to medical journalism and research – including over-simplification, sensationalism, attribution bias, unjustified mechanisms, inadequate sampling, emphasis on surrogate outcomes, and inattention to comparative effectiveness. Clinicians and researchers must be vigilant to avoid falling victim to the "Swift effect," since failure to scrutinize available evidence can lead to acceptance of unjustified theories and negatively impact clinical decision-making.

**Data availability statement:** The full dataset and statistical code used for this study is available as an online supporting information. The are also publicly available at Open Science Framework's (OSF) depository, at https://osf.io/fmney/.

**Funding:** The author(s) received no specific funding for this work.

**Competing interests:** The authors have declared that no competing interests exist.

## Introduction

The love story between pop icon Taylor Swift and National Football League (NFL) star Travis Kelce not only captured headlines, but sparked discourse among sports analysis and pop culture fans alike. A captivating theory emerged that Swift's presence at Kansas City Chiefs games boosted Kelce's on-field performance. Sports enthusiasts speculated that if Swift attended, Kelce would be a better man on the field and the Chiefs would win; conversely, her absence was linked to team defeat [1–3]. This perceived causal relationship became known as the "Swift effect" [4,5]. Financial stakes were likely influenced by Swift's game attendance, with substantial increases in the number of bets favoring Kelce [6–8] – leaving fans haunted by huge losses [7,9].

However, the validity of the "Swift effect" is questionable, as it is rooted in limited, short-term observations and superficial statistical analyses that fail to consider the myriad of factors which influence football performance. Moreover, the purported performance boost of Swift overlooks critical context – Kelce's and the Chiefs success predates Swift's entry into the NFL scene [10]. The rush to attribute causality to Swift's attendance is nothing new. Rather, the "Swift effect" epitomizes a broader issue prevalent in mainstream media communication and healthcare practice: compelling narratives often overshadow strong evidence, potentially leading to misguided decision-making [11–13]. Examples of these widely cited poorly evidenced, such as high temperature and relative humidity significantly reduce spread of COVID-19 [14], shark cartilage cures cancer [15], and that chocolate milk could protect the brain from sports concussions [16].

Thus, we designed a study to critically evaluate the "Swift effect" using robust statistical methodology. Specifically, we aimed to examine if: 1) Swift's presence boosted Kelce's performance, and 2) Swift's absence impaired Kelce's performance. Secondarily, we aimed to determine if Swift's attendance status influenced the Chiefs' game outcomes. Through dissecting the "Swift effect," we seek to emphasize the importance of appropriate reasoning and statistical analyses in shaping informed decisions, drawing parallels to medical research and clinical practice.

## Methods

### Study overview

To assess whether Taylor Swift's attendance actually influences NFL game performance, we designed a comprehensive quasi-experimental study comparing Swift-era games (2023 season) with a matched set of historical games. Central to our approach was the use of covariate matching, whereby we used historical data to estimate what Kelce's expected performance would be using data-driven pre-game predictors (i.e., pre-game Elo ratings). This approach enabled us to create well-balanced "treatment" (Swift-era data) and "control" (pre-Swift era historical data), effectively reducing potential confounding and accounting for natural performance variability. Subsequent analyses using linear mixed effects and logistic regression models allowed us to rigorously test for performance differences attributable to Swift's

attendance, to ultimately determine if any apparent "Swift effect" was statistically robust. Full details of each step are described below.

## Experimental design

It was not possible to design a randomized clinical trial in which Swift was assigned to initiate a romantic relationship with Kelce at a specified time, nor to terminate this dating intervention at pre-determined time. Since we did not have financial resources to offer her an honorarium, we did not feel comfortable asking Swift to alter her game attendance schedule for the purposes of our scientific research. Instead, we used a quasi-experimental design using covariate matching to answer our research questions.

This covariate matching approach lacks the causal certainty of a randomized trial, but provides more robust insight than an uncontrolled observational or quasi-experimental study by attempting to control for confounding factors that could influence Kelce's performance [17,18]. We aimed to isolate the so-called "Swift effect" by comparing Kelce's performances of the 2023 season (Swift era), when Swift was either cheering from the bleachers or notably absent, to his historical performances (pre-Swift era) in comparable games. This allowed us to differentiate whether variation in Kelce's performance and the Chief's record in the 2023 season were likely attributable to the presence or absence of Swift, or simply reflected typical variability inherent to team sports.

## Data collection

All data were publicly available. Travis Kelce's performance metrics and game outcomes were obtained from the ProFootballReference database [19]. These included all metrics associated with Kelce's performance (e.g., total yards, yards per catch, number of touchdowns, fumbles, blocks, etc.) as well as the Chief's overall performance (win-or-loss status, point scored, etc.). Based on media claims of the "Swift effect," [1–3] we focused our analyses on Kelce's total yards and Chiefs win-loss status.

Weekly team Elo ratings were obtained from an publicly available online source [20]. Taylor Swift's presence or absence at games was obtained from online news sources.

## Data categorization

We divided the dataset into two distinct eras – pre-Swift (2014–2022 seasons) and Swift (2023 season). This division was necessary for multiple aspects of our statistical analysis, described below. We did not collect or analyze data from the 2024 season, since this manuscript was first submitted before that season began.

## Statistical analysis

All analysis was performed in R version 4.2.2. All code and relevant data is available as Supporting Information (S1 File – Dataset; S2 File – Statistical Analysis Code).

**Swift effect on kelce's performance.** To explore the potential impact of Taylor Swift's presence on Travis Kelce's performance, we utilized a robust quasi-experimental framework, with a three step process: 1) identify confounding factors that could influence Kelce's performance, 2) control for confounders using covariate matching, 3) evaluate the effect of Swift's game attendance using linear mixed effects modeling. Details of each step in the process are described below.

Step 1 – Identifying Confounding Factors

We sought to determine which confounding factors influenced Kelce's performance, so we could control for them using covariate matching. For this, we examined a number of metrics related to Elo scores. Elo scores were originally developed for ranking chess players, but are now used for rating/ranking in various sports. The Elo score is derived from an algorithm commonly used in sports analytics to quantify overall team strength [21]. The algorithm considers previous game outcomes (including magnitude of victory or loss), as well as the Elo score of the opposing team (i.e., defeating an opponent

with a high Elo will boost one's Elo score more than defeating an opponent with a low Elo score). A typical Elo score for an NFL team ranges between 1200 and 1800, and is updated following each game. Differences between team Elo scores can be used to compute the probability of a given team's win or loss.

To identify the most relevant confounding factor(s) [i.e., the strongest predictor(s) of Kelce's performance in the pre-Swift era], we used linear regression with stepwise selection. Kelce's yardage was the dependent outcome. Chief's pre-game Elo, opponent pre-game Elo, Chief's win probability, and game location (Chief's home vs. away) served as independent predictor candidates. We found Chief's pre-game Elo score was the best predictor of Kelce's performance (Table 1, and further described in Results section).

Step 2 – Covariate Matching

Each game from the Swift era was matched to five corresponding games from the pre-Swift era, based on closely aligned Elo scores. For example, the Chiefs had a pre-game Elo of 1678.1 in Game 3 of the 2023 season (Swift-era), and this game was matched to five other games from other pre-Swift seasons (with an Elo range of 1675.3 to 1680.0). Thus, each of the six games within each "matched set" represented similar pre-game situations, reducing the role of a key confounder (team strength, as quantified by Elo) and facilitating a direct comparison between the pre-Swift versus Swift eras.

The 5:1 matching ratio was selected *a priori*, with the justification that it allowed a balance between having multiple "control" games from the pre-Swift era, without sacrificing the quality of the match (i.e., all matched games had Elo scores within a narrow range). Likewise, the use of multiple games increases the sample size and allows one to examine the natural variability in performance that could be expected between different games. However, we also conducted a *post hoc* sensitivity analysis to determine how different matching ratios (1:1, 2:1, 3:1, 5:1, and 8:1) and methods (non-replacement versus with replacement) influenced the results. We selected these specific values as a reasonable representation of how small and large matching ratios would influence the results, if at all.

The covariate matching algorithm used the R library (matchit) and employed the "optimal" matching method. This minimizes the total difference between matched pairs. All possible pairings are considered and those with the lowest total distance are used. Our *a priori* 5:1 match algorithm used non-replacement, such that once a game from the Pre-Swift era was matched to another game from the Swift era, it was no longer available to be matched with any other games.

Step 3 – Analyzing Performance Differences

Linear mixed effects models were developed to compare Kelce's performance between the two eras. For all models, Kelce's performance, quantified as total yards per game, was the dependent variable. While a tight end's overall performance includes various other metrics (e.g., catch rate, blocking, fumbles, etc.), we focused on yardage for two key reasons: 1) it has been the primary metric highlighted in popular discourse, and 2) as a continuous variable with a large range, it was statistically convenient to examine using linear modeling.

Matched set (i.e., a Swift era game matched to five pre-Swift era games) was included as a random effect to account for intra-group correlation within each cluster of matched games. Pre-game Elo was included as a covariate to adjust for team strength. Era (Swift vs. Pre-Swift), served as a categorical fixed factor. We explored multiple different covariance structures to optimize model fit and selected the model with the lowest Akaike information criterion (AIC) as the final model.

Since Swift was not present for any of the pre-Swift era games, it was not appropriate to create a single model with Swift's presence/ absence serving as a fixed factor or explore a season by Swift attendance interaction (i.e., the dataset

**Table 1. Results of linear effects models examining factors which influenced Travis Kelce's performance in the Pre-Swift era (2014-2022).** Chiefs' pre-game Elo was the only predictor that was entered into the model. Opponent Elo score, Chief's win probability, and home/away game status were not selected for inclusion in the final model.

| Intercept (95% CI) [p-value] | Pre-game Elo (95% CI.) [p-value] |
| --- | --- |
| −172.8 (−316.9, −28.7) [0.019] | 0.15 (0.06, 0.24) [<0.001] |

would be unbalanced). As such, two separate models were constructed. The first model aimed to determine if Swift's presence in 2023 influenced Kelce's performance, compared to what would be expected compared to games from the Pre-Swift era. The second model aimed to determine if Swift's absence in 2023 influenced Kelce's performance, compared to what would be expected.

**Effect on chiefs game outcomes.** To examine the role of Swift's attendance on the Chiefs' game outcomes, we used mixed-effects binary logistic regression. Game outcome (win or loss) was the dependent variable. Matched set was included as a random effect. Swift's attendance status (present or absent) was entered as a binary predictor variable. Multiple models were fit with different combinations of pre-game Elo-based metrics as continuous predictor variables, and the model with the lowest AIC was selected as the final model.

### Patient and public involvement

This study was developed following the authors' conversations about the validity of the "Swift effect" with multiple members of the general public, including Swifties, football fans, and sports betting enthusiasts. Prior to submission, the manuscript was reviewed by a 13-year old female Swiftie and her mother (both unrelated to the authors and unaffiliated with the medical community).

## Results

### Game information

We identified a total of 180 NFL games that Travis Kelce participated in. Of these, 19 were in the 2023 season, and Swift attended 13 (68.4%) of these games.

The Chiefs won 10 of the 13 games (76.9%) that Swift attended, and 4 of the 6 games (66.7%) that Swift did not attend.

### Kelce performance summary

For the 2023 season, Kelce achieved a mean (standard deviation) of 79.9 (42.8) yards when Swift was present and 50.0 (28.1) yards when she was absent. Overall, Kelce's performance in the 2023 season was 70.5 (40.5) yards per game (n = 19 games), compared to 73.9 (36.7) yards per game in the pre-Swift era (n = 161 games).

### Confounding variables and covariate matching

The stepwise selection process selected the Chiefs' pregame Elo score as the only predictor of Kelce's performance. Regression parameters from the final model are presented in Table 1. Based on the model, Kelce would be expected to achieve, on average, approximately 15 more yards for every 100 point increase in the Chiefs pre-game Elo score. Since opponent pre-game Elo and Chiefs' win probability were not included in the model, this suggests that Kelce's performance is most dependent on the Chief's strength as a team, and not the relative strengths of the opposing team. The $R^2$ and adjusted $R^2$ for this model were 0.067 and 0.061, respectively, which indicates that most (~94%) of the variability in his performance is due to other factors, which we could not account for. When the next-best predictor (opponent elo score) was forced into the model, the $R^2$ only improved to 0.069 and adjusted $R^2$ deteriorated to 0.057.

Regardless, the significant effect for pre-game Elo does indicate it is a potential confounding factor that should be controlled for.

Fig 1 shows the results of covariate matching. Following our *a priori* covariate matching procedure (5:1 non-replacement approach), the standardized mean differences for pre-game Elo ratings were reduced from 0.76 to below 0.01, confirming that all Swift-era games were successfully matched with comparable historical controls. This quantitative evidence supports the assertion that the treated and control groups achieved acceptable covariate balance.

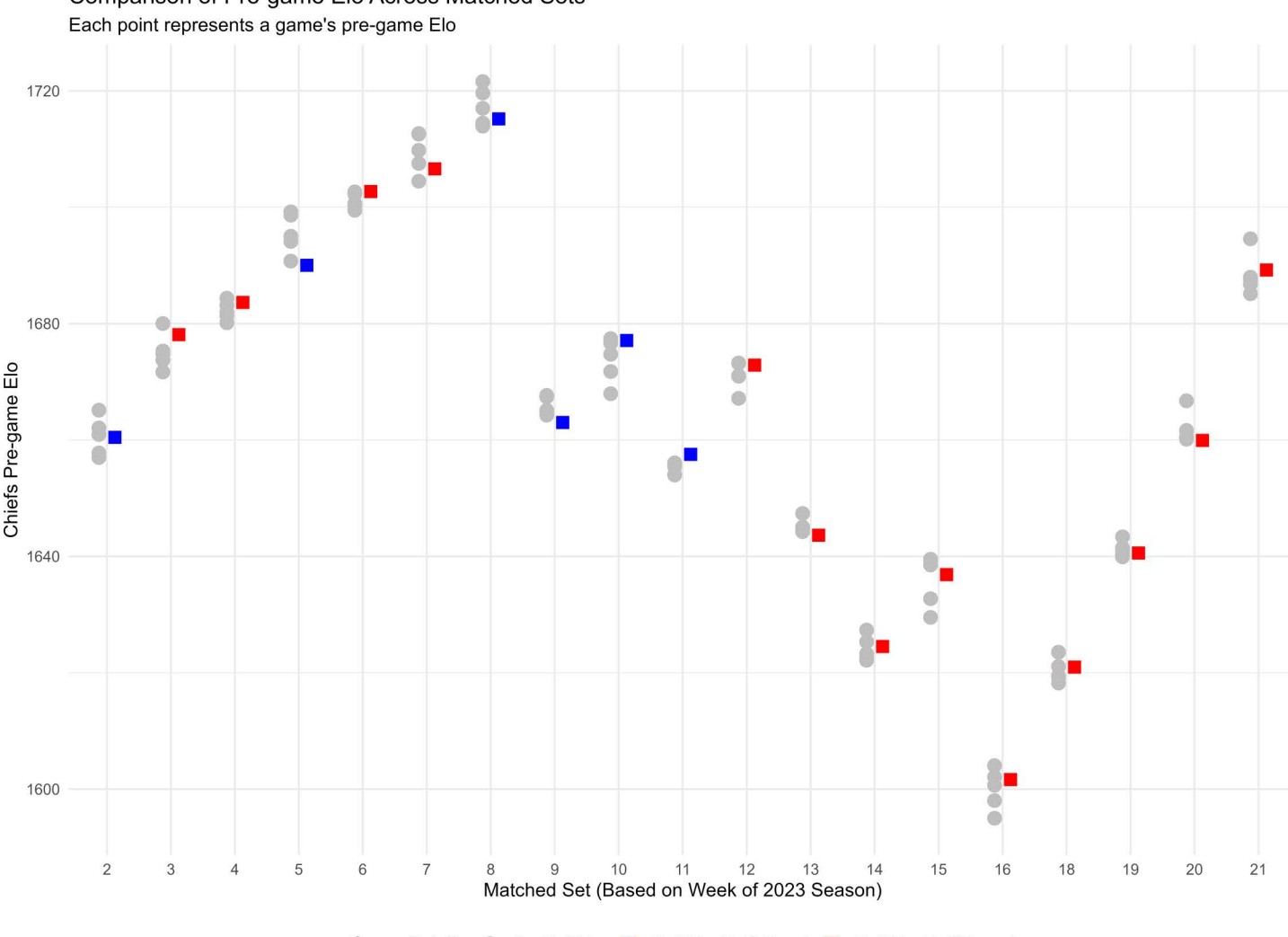

**Fig 1. Matching Pre-Swift and Swift era games.** Each vertical column of data points along the x-axis represents one "matched set", consisting of one game from the Swift era (2023 season, colored squares), and five games from the pre-Swift era (2014-2022, grey circles). Red squares indicate games that Swift attended in the 2023 season, and blue square represent games she did not attend in the 2023 season. The y-axis represents the Chiefs's pre-game Elo value, which quantifies team's strength (higher values indicate better performance) and is also related to Kelce's performance (see S3 File – Supplemental Results, for details regarding Elo). The similar pregame Elo values within each matched set demonstrates the matching procedure was effective. This allows Kelce's Swift era performances to be compared to similar games from the pre-Swift era. Note: Kelce did not play in Week 1 or 17 of the season, thus these numbers are missing from the x-axis.

## Swift effect models

   **Effect on Kelce.** A summary of model parameters is presented in Table 2. For each model, the best fit was achieved using a first-order autoregressive covariance structure.

   The first model examined whether Swift's presence at games during the 2023 season had a performance enhancing effect on Kelce. When Swift was present (n = 13 games in the 2023 season), Kelce achieved 7.1 [95% confidence interval: (−12.7, 26.9)] yards per game more in the Swift-era than he did in matched games from previous seasons (n = 65 games), but this was not statistically significant (p = 0.476).

**Table 2. Results of linear effects models examining the Swift effect on Travis Kelce's performance.** Taylor Swift's presence or absence did not have a statistically significant effect on Travis Kelce's performance during the Swift era, relative to Elo-matched games from the pre-Swift era.

| Model | Intercept (95% confidence interval) [p-value] | Eras (95% confidence interval) [p-value] | Pre-Game Elo (95% confidence interval) [p-value] |
|---|---|---|---|
| Swift Present | −131.1 (−450.1, 187.8) [0.414] | 7.1 (−12.7, 26.9) [0.476] | 0.123 (0.12, 0.32) [0.204] |
| Swift Absent | 116.2 (−1201.4, 1433.8) [0.858] | −28.6 (−69.4, 12.3) [0.163] | −0.023 (−0.81, 0.76) [0.954] |

The second model examined whether Swift's absence from games influenced Kelce's performance. When Swift was absent (n = 6 games in the 2023 season), Kelce achieved 28.6 (95% CI: −69.4, 12.3) yards fewer than he did in matched games in previous seasons (n = 30 games) – however, this difference was also not statistically significant (p = 0.163).

Fig 2 shows how Kelce's performance compares between the Swift era and pre-Swift era.

**Effect on chiefs.** The best-fitting logistic regression model included Chiefs' pre-game win probability in the model. Swift's presence was not related to the Chiefs' game outcome (p = 0.692), nor was win probability (p = 0.149). Model parameters are presented in Table 3. Data from the sensitivity analysis revealed the results were relatively stable (all non-significant, with large confidence intervals spanning 1.0) and are presented in Supporting Information (Table S2) in S3 File.

**Sensitivity analysis.** Across a range of matching specifications (from 1:1–8:1 matching ratios), with and without replacement, the estimated effect of Swift presence remains close to zero and statistically non-significant (range: −3.3 to 7.1 yards). The effect of Swift's absence was consistently negative, but varied widely by matching method (range: -11.9 to -45.3 yards).These data are presented in Fig 3, with further detail in the Supporting Information (Table S1) in S3 File. While the estimated effect of Pre-Game Elo is consistently positive and significant in several models (particularly in the non-replacement specifications), its magnitude tends to diminish in models using replacement matching or higher matching ratios. Overall, despite some variability in intercepts and the precise value of the Elo coefficient, the main substantive conclusion—that Swift's attendance does not produce a meaningful change in performance—remains robust.

## Discussion

Despite widespread claims that Swift's attendance directly contributed to better outcomes for Kelce and the Chiefs [1,2,5], our analysis revealed that Kelce's performances with Swift present closely adhered to his historical averages. Limited data could suggest that Kelce's performance was impaired when Swift was not present (generally at a non-significant level, but dependent on which matching algorithm was used), but the small sample size (n = 6 games) precludes any definitive conclusions. Likewise, there was no statistical evidence that Swift influenced game outcomes. Thus, sports bets based on Swift's attendance are rooted in folklore, rather than science. This epiphany is surprisingly relevant to the healthcare community, in that the "Swift effect" shares many unfortunate parallels to scientific communication and medical research.

### Sensationalism and over-simplification are treacherous

Our study's findings starkly contrast the narrative perpetuated by mainstream media [1–3], where the "Swift effect" was not only sensationalized, but also oversimplified into a seemingly clear-cut causal relationship. The media, much like a catchy pop song, often gravitates towards stories that are easily digestible and entertaining, even if they don't full align with the complexities of reality. Journalistic attempts to explain medical research to non-clinicians through simplification and drawing parallels to more familiar topics may result in further distortion of scientific accuracy, and ultimately media

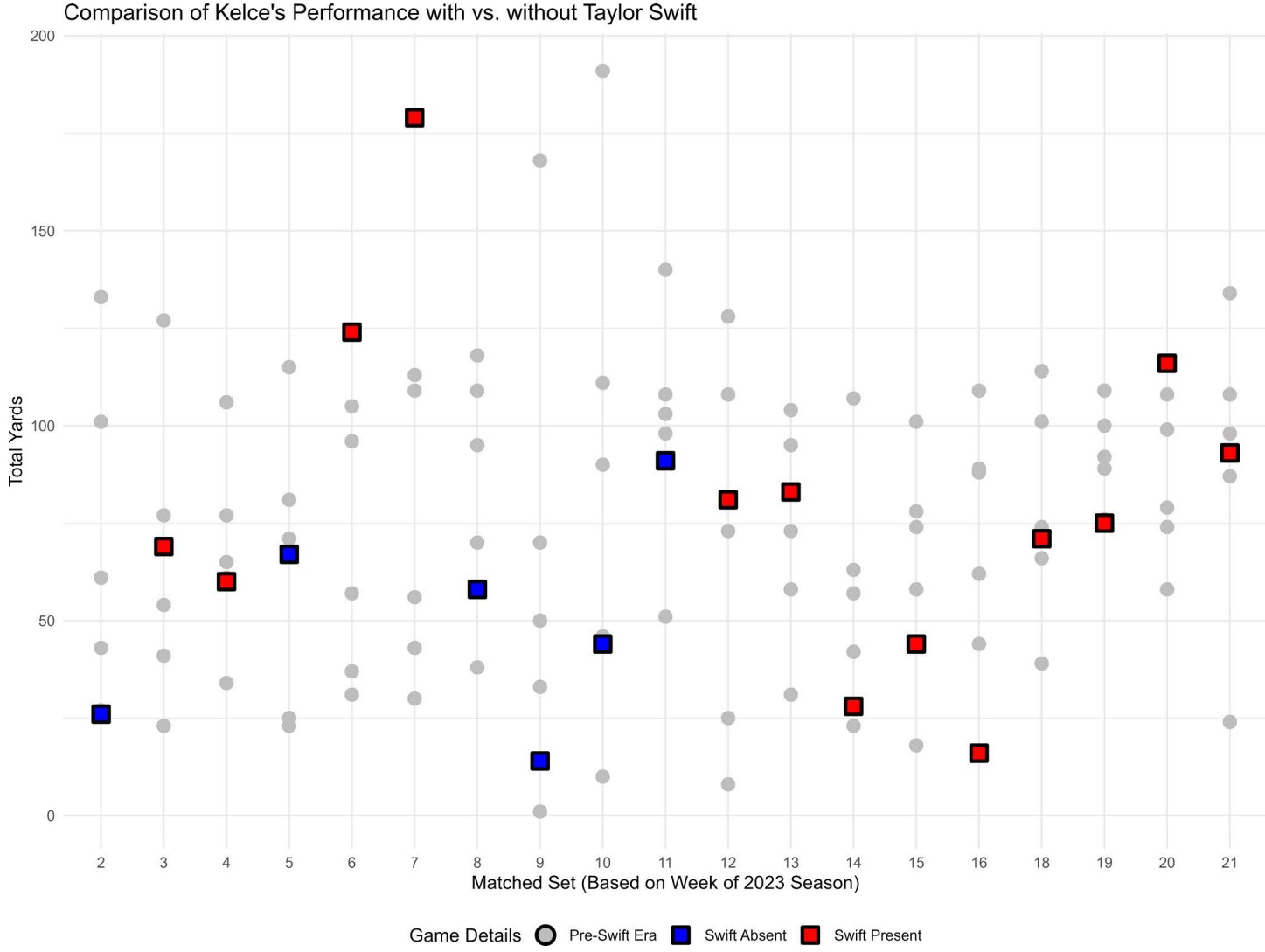

**Fig 2. Comparison of Kelce's performances between Pre-Swift and Swift eras.** Each vertical column of data points represents one "matched set", consisting of one game from the Swift era (2023 season, colored squares), and five Elo-matched games from the pre-Swift era (2014-2022, grey circles). Red squares indicate games that Swift attended in the 2023 season, and blue square represent games she did not attend in the 2023 season. The y-axis represents Kelce's yards per game (higher values indicate better performance). There is substantial variability in Kelce's game performances in both eras. When Swift is present, there are examples of him playing unusually well (i.e., Week 7) and unusually poor (e.g., Week 16). Note: Kelce did not play in Week 1 or 17 of the season, thus these numbers are missing from the x-axis.

**Table 3. Results of binary logistic regression model examining the Swift effect on Chiefs' game outcomes.** Parameters are exponentiated (i.e., $\exp(\beta)$) to achieve odds ratios. Taylor Swift's presence in the 2023 season did not have a statistically significant on the Chiefs likelihood of a victory. The best model fit included Chief's pregame win probability as a continuous variable in the model, but this was also not statistically significant.

| Swift Attendance (0 = Absent, 1 = Present) (95% confidence interval) [p-value] | Chiefs Win Probability (95% confidence interval) [p-value] |
|---|---|
| 1.32 (0.33, 5.34) [p=0.692] | 9.83 (0.44, 220.9) [p=0.149] |

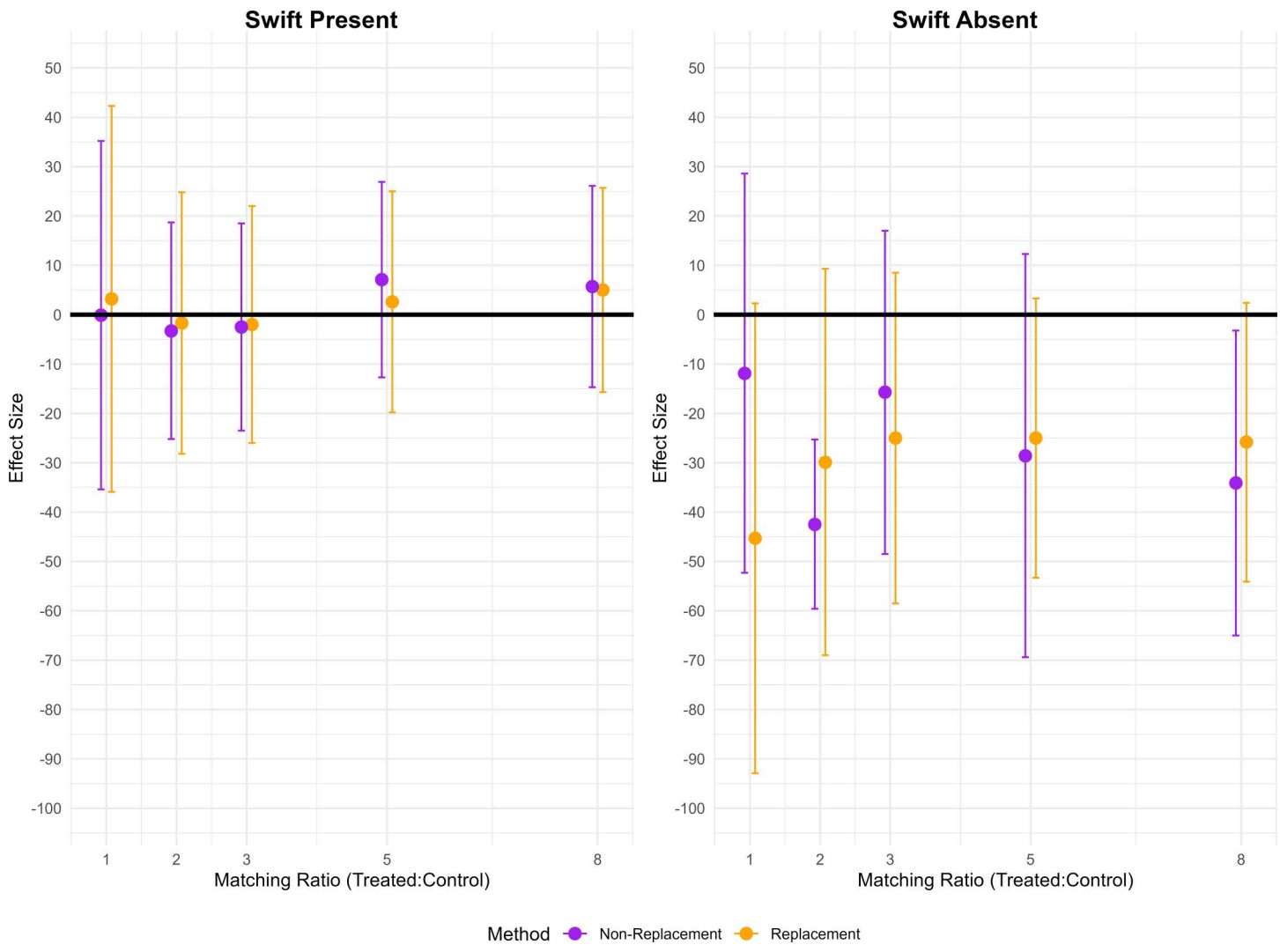

**Fig 3. Sensitivity Analysis of the "Swift Effect" Across Matching Algorithms.** Panel A (left) shows results for games in which Taylor Swift was present, and Panel B (right) shows results for games in which she was absent. On the x-axis, the matching ratio (1:1, 2:1, 3:1, 5:1, and 8:1) is plotted on a continuous scale. For each matching ratio, results from two matching methods are displayed: replacement (orange markers) and non-replacement (purple markers). The y-axis represents the estimated effect in Kelce's yards (with corresponding 95% confidence intervals shown as error bars) from linear mixed effects models comparing Swift-era games to matched pre-Swift controls. In Panel A, the effects indicate that Kelce's performance does not differ significantly when Swift is present, while Panel B reveals that any performance decline when she is absent is sensitive to the matching algorithm employed. These findings illustrate that, despite some minor variability in point estimates and confidence intervals across different matching specifications, the overall conclusion regarding the lack of a robust "Swift effect" remains stable.

sensationalism [22]. This reductionist approach can lead to widespread misinterpretation of the true effects (or lack thereof), as the nuances and limitation of the underlying data are glossed over in favor of a more compelling storyline.

Such sensationalism is not only limited to the media, but also researchers themselves – lexicographic analysis reveals hyperbole and exaggeration in reporting results, likely in an effort to market research for publication [23]. Likewise, retrospective review found that approximately 35–40% of academic-associated press releases on health-related topics contained exaggerations (i.e., correlation equals causation, animal data extrapolated to humans), and these exaggerations significantly increased the likelihood of exaggeration being present in the media [24]. Low-quality studies may

masquerade as groundbreaking medical breakthroughs, just as basic science findings may be extrapolated into clinical advice. Much as the "Swift effect" seemingly caused fans to bet large sums of money based on flawed assumptions, clinical decisions can be misguided by overhyped but inadequately supported medical research. Indeed, negative consequences have been observed across a gamut of medical specialties [25–29].

## Lessons in attribution and confirmation bias

The investigation into the so-called "Swift effect" may provide an easily relatable example of how captivating narratives often unravel when subjected to rigorous statistical scrutiny. Our findings also exemplify various biases which can inappropriately lead to inferring causality. While Kelce exhibited peak performances during earlier weeks of Swift's attendance (e.g., the fortnight period of weeks 6 and 7), similar performances were observed in the pre-Swift era. This suggests these particular games were within the bounds of expected variability. Nonetheless the desire to connect Swift's presence with these outcomes exemplifies attribution bias. [30] Likewise, after his season-best performance in Week 7, Kelce's yardage declined substantially in the following game. This drop might be explained by strategic adjustments from opposing teams (e.g., defense focused on stopping the big star) or simple regression to the mean (which is well-described in sports [11,31]). However, this coincidental alignment with Swift's absence perpetuated confirmation bias [32] – fueling the narrative that Kelce performs better when she is present. This led to her attendance plans being kept top secret in an effort to limit her influence on betting odds [3].

Stakes in Super Bowl betting may not be critically important for society, but the phenomenon of making decisions, including financial investment and resource allocation, based on unsupported claims is not confined to the realm of sports. For example, Goldacre describes how exaggerated claims about drug efficacy, often disseminated through press releases and media reports, have led to significant misallocations of resources in the pharmaceutical industry and premature adoption of therapies that ultimately fail under rigorous testing [11,12]. Likewise, low reproducibility rates in preclinical research is estimated to lead to>US$28 billion per year in research waste [33]. Another example can be seen in Wakefield's initial claims linking the MMR vaccine to autism – although the initial claims were later found to be false (and based on fraudulent data), they still had direct impacts on society (i.e., measles outbreaks) [34,35]. These examples underscore that reliance on preliminary, incomplete, or non-rigorous evidence is not unique to sports, but a pervasive problem that can lead to misguided decisions. Our study is intended to serve as a countermeasure example, ensuring that any observed associations, such as the purported "Swift effect" are rigorously tested and not merely the product of selective or oversimplified data interpretation.

## Parallels to medical research

The "Swift effect" offers several parallels to issues in medical research and clinical practice. Kelce's initial performance "improvement" is similar to many short-term studies which show promising results (which may be due to direct and/or indirect effects [36]). Kelce's sub-par performance later in the season (e.g., Weeks 14 and 16) could even be compared to an undesirable side-effect of Swift's attendance which was not recognized early on. Just as Kelce's performance regressed toward the mean, early medical interventions might show initial efficacy that diminishes over time, emphasizing the need for extended follow-up in clinical trials. Furthermore, the limited number of games Swift attended, and those she did not, reflect the challenges of underpowered studies in medical research. This consequence of small sample size are made clear by our sensitivity analysis, which showed fragility in the results for this small subset of games that she did not attend (i.e., some matching methods produced statistically significant results, whereas most did not). Small sample sizes can produce false positives or negatives, leading to distorted conclusions [37–39]. For instance, a single center study (n = 1548) by Van den Berghe et al. reported suggesting intensive insulin therapy reduced 12-month mortality [40], yet the larger, multicenter NICE-SUGAR trial (n = 6104) demonstrated increased 90-day mortality with intensive glucose control [41], highlighting how sample size and study design can lead to drastically different and more reliable clinical conclusions.

The focus on Kelce's yards per game as a performance metric is akin to relying on surrogate outcomes in clinical trials. In this case, the most important outcome (Chiefs win or loss) is known, so a surrogate (Kelce's individual performance)

is largely unnecessary. Much like Kelce's yardage is only one aspect of the Chiefs' overall game outcome, surrogate outcomes often provide an incomplete picture of the response to a clinical intervention. Overemphasis on surrogate outcomes has led to major trial failures (with substantial financial waste) and poor clinical outcomes [42–45]. A classic example comes from the CAST study, in which myocardial infarction survivors were proactively treated with antiarrhythmic agents to reduce/prevent premature ventricular contractions (surrogate outcome, indicative of cardiac risk), yet were actually found to have a substantially increased risk of death – causing the trial to be discontinued [35].

Claims of causality should be rooted in a solid mechanism [46] – the notion that the presence of one specific spectator (within an audience of 60,000 + attendees) could consistently influence an entire team's performance is a foolish one. Similarly, some therapeutic interventions are rooted in weak scientific evidence [47] (or the perpetuation of research fraud in a vulnerable peer-review system [48]) and have an implausible mechanism. As an example, the Q-collar jugular compression device claims to "protect the brain" from sports impact by mimicking the woodpecker's natural physiology and the body's response to "higher altitudes" [49]. Yet, the device failed to reduce concussion risk in athletes [50] – likely because the woodpecker [51] and altitude [52–54] justifications turned out to be unsubstantiated. Rigorous preclinical validation and independent replication is needed to ensure that new interventions are safe & sound – this is essential to avoid misallocation of financial resources and ensure that patients are not harmed.

Finally, the focus on Swift's role in the 2023 season overlooks the Chiefs' and Kelce's prior successes, drawing a parallel to the issue of comparative efficacy. Proponents of the "Swift effect" did not compare the current intervention (i.e., Chiefs coaching + Swift/Kelce romance) to the existing treatment (i.e., Chiefs' coaching alone). Chiefs fans who believe in the "Swift effect" may wish for Swift to be present evermore, yet the season outcome remained unchanged in her presence (i.e., Chiefs SuperBowl wins in both, the pre-Swift and Swift eras). Evaluating new treatments should always consider existing standards to contextualize their true value [55].

## Limitations

This study was performed to examine the media narrative [1–3] regarding a causal effect of Swift on Kelce and the Chiefs – the implausibility of the hypothesis could itself be considered a limitation. Although there is little scientific justification for studying this topic, claims about the "Swift effect" have received substantial media attention and may have influenced sports betting decision – thus, it is important for scientists to engage with the public and address misconceptions on this topic.

Given that the intent of this study were to examine the validity of media claims, we only examined the two most reported outcomes associated with the "Swift effect" (Kelce's yards and Chief's win-loss status) [1–3]. However, these may not fully reflect all aspects of performance. For instance, our analyses do not account for how well Kelce performed in catching the ball as an intended pass target (e.g., receiving percentage), nor did it analyze his performance in other duties (e.g., blocking) or if he made errors (i.e., fumbles). In addition to not being as widely scrutinized in "Swift effect" stories, they are discrete data and sometimes zero-inflated – as such, mathematical models examining these would be less straightforward to interpret and potentially more prone to error due to narrower data distributions.

Elo was the only metric identified as a potential confounding factor that could be effectively balanced using our matching procedure. While this allowed us to reduce bias in our comparisons, it does not capture all aspects of performance. Kelce's on-field metrics are influenced by various factors, including his game day physical capabilities and emotional state, Chief's strategies, opponent's strategies, and all players' effectiveness in carrying out their respective strategies. These important factors are difficult to quantify and incorporate into a matching framework.

This study assumes that Kelce's performance, and the relationship between his performance and team Elo rating have remained constant over the course of his career. Age-related changes in American-style football performance may be difficult to quantity, since they are highly dependent on team dynamics and this itself may change from year-to-year, or even within a season (e.g., based on player injuries, coaching, etc.) Matching 2023 season games to all games from Kelce's

NFL career does not account for any of this variation. Regardless, our preliminary analysis did identify Elo rating throughout the pre-Swift era to be related to his performance, and there is no evidence that this relationship had any systematic change which would introduce bias into our analysis.

It is possible that other factors, such as opposing team defense could also influence the primary outcome variable (Kelce's yardage) and our analysis does not capture this. While there are various metrics to quantify team defense (e.g., points against, yards against, etc), they do not necessarily relate to how a team handles one specific player. We explored whether the opposing team's Elo rating (an overall metric of team performance, not just defense) influenced Kelce's or the Chief's performance, but stepwise regression models suggested that it did not. Therefore, we felt there was not sufficient justification to include it in our analysis. Likewise, the use stepwise entry selection of variables has its own limitations, however we felt the candidate variables were appropriately justified. Given the nature of this study was to examine the validity of a media-fueled claim about one specific celebrity couple, the research methods and statistical analyses performed should be sufficient, despite the presence of some limitations.

In addition to the real limitations, we could facetiously suggest that our model was too simplistic to capture the influence of Swift's role in Kelce's performance. Kelce is merely one of 22 players on the field at a given time, and it is improbable that his performance is uniquely susceptible to the presence or absence of a significant other. Therefore, it is plausible that Kelce's performance metrics are influenced by a complex interplay of factors beyond Swift's attendance, potentially including the romantic entanglements of the other 21 players. Our model did not account for the attendance of girlfriends, spouses, or hypothetical mistresses from illicit affairs from all 22 players, but future studies should utilize quantum computing to analyze a model with 4,194,303 terms to examine this (i.e., 22 fixed factors, 231 two-way interactions, 1540 three-way interactions, etc.). There is also some possibility that other factors, such as team strategy and player skill, exert a more substantial impact on performance metrics than the presence or absence of specific spectators, such as Swift.

## Conclusions

Our analysis provides the answer Swift and Kelce fans have been looking for, which has been here the whole time – the "Swift effect" appears to have little influence on Kelce's performance or the Chief's end game. Long story short, Kelce did not run the ball any better with Swift present than he did in previous seasons. Call it what you want – a hoax, or an innocent mistake in statistical analysis, but the "Swift effect" is not merely a case of the general public being susceptible to compelling narratives (and losing money in sports betting). Rather, the "Swift effect" epitomizes a constellation of issues which permeate medical news and peer-review literature, including sensationalism, attribution bias, inadequate sample sizes, insufficient follow-up periods, unjustified mechanisms, excess focus on surrogate outcomes, and inattention to comparative effectiveness. Thus, clinicians and researchers must remain vigilant in scrutinizing the validity and quality of evidence underpinning drug development, clinical trials, and patient care to avoid falling victim to their own version of the "Swift effect" – where the stakes are far greater than that of any football game.

## Supporting information

**S1 File. Dataset.** This is the dataset used for the study.
(XLSX)

**S2 File. Statistical Analysis Code.** This is the R code used for data analysis.
(TXT)

**S3 File. Supplement Results.** This includes two supplemental tables: **Table S1.** Sensitivity analysis for models examining the effects of Swifts presence or absence on Travis Kelce's performance. **Table S2.** Sensitivity analysis for models examining the effects of Swifts presence or absence on Kansas City Chief's win probability.
(DOCX)

## Author contributions

**Conceptualization:** James M. Smoliga.

**Data curation:** Kathryn E. Sawyer.

**Formal analysis:** James M. Smoliga.

**Investigation:** James M. Smoliga, Kathryn E. Sawyer.

**Methodology:** James M. Smoliga.

**Project administration:** Kathryn E. Sawyer.

**Software:** James M. Smoliga.

**Visualization:** James M. Smoliga.

**Writing – original draft:** Kathryn E. Sawyer.

**Writing – review & editing:** James M. Smoliga, Kathryn E. Sawyer.

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
