## [Decision Letter · Decision Letter 0]

3 Jan 2025

PONE-D-24-37729Fact or folklore? A Quasi-Experimental Investigation into Taylor Swift's Influence on American football performancePLOS ONE

Dear Dr. Smoliga,

Thank you for submitting your manuscript to PLOS ONE. After careful consideration, we feel that it has merit but does not fully meet PLOS ONE’s publication criteria as it currently stands. Therefore, we invite you to submit a revised version of the manuscript that addresses the points raised during the review process.

We look forward to receiving your revised manuscript.

Kind regards,

Mohsin Shahzad

Academic Editor

PLOS ONE

2. Thank you for uploading your study's underlying data set. Unfortunately, the repository you have noted in your Data Availability statement does not qualify as an acceptable data repository according to PLOS's standards. At this time, please upload the minimal data set necessary to replicate your study's findings to a stable, public repository (such as figshare or Dryad) and provide us with the relevant URLs, DOIs, or accession numbers that may be used to access these data. For a list of recommended repositories and additional information on PLOS standards for data deposition, please see https://journals.plos.org/plosone/s/recommended-repositories.  (JNC (Straive) 26 Nov 2024: Potentially unacceptable repository)

3. Please include a copy of Table 1 which you refer to in your text on page 12.

Additional Editor Comments (if provided):

Reviewers' comments:

Reviewer's Responses to Questions

**Comments to the Author**

1. Is the manuscript technically sound, and do the data support the conclusions?

Reviewer #1: Yes

Reviewer #2: Yes

2. Has the statistical analysis been performed appropriately and rigorously? 

Reviewer #1: Yes

Reviewer #2: No

3. Have the authors made all data underlying the findings in their manuscript fully available?

Reviewer #1: Yes

Reviewer #2: Yes

4. Is the manuscript presented in an intelligible fashion and written in standard English?

Reviewer #1: Yes

Reviewer #2: Yes

5. Review Comments to the Author

Reviewer #1: GENERAL COMMENTS

Thanks for the opportunity to read and review this paper. I quite enjoy the notion of using pop culture references alongside a little bit of a satirical take to highlight the importance of appropriate scientific communication and analysis of data, while also working in sports science myself – so I’d say this paper sits right in a niche ally of interest for me. Given the nature of this paper, I haven’t necessarily been overly critical in my review of the science, but in saying that this is a well written and conducted scientific study, albeit of a topic not overly important at face level. I think the premise of this paper is highlighted quite well in the authors approach, but there are opportunities to be more explicit (see specific comments on title and potential need for case study examples from medicine and health at different points). I’ve also left some detailed comments on how certain choices in the paper and communication of methodology is perhaps a little on the light side, and hence falls into a different but also important trap in scientific writing and research (i.e. isolated focus on individual outcomes, detail in replicating methods). Nonetheless, this is an enjoyable and easy read that uses a unique approach in highlighting a relevant problem to scientific communication to the media.

Specific comments on each section are provided below.

I recently received a review on one of my own papers where the reviewer provided an ‘about me’ section, which I found quite valuable in placing additional context around their comments, so it is something that I am trying to add to my own reviews:

I have a research background primarily in sports injury and performance biomechanics, but have a secondary research focus on using data and analytics approaches to characterise and improve sports performance. My work in this area has predominantly focused on netball, with a little in Australian football.

I am also open to being identified on papers I reviewed – so am signing off my review here.

Aaron Fox

Centre for Sport Research

Institute for Physical Activity and Nutrition

Deakin University

SPECIFIC COMMENTS

Title

I think the title should incorporate some element that highlights the broad notion that this paper is trying to highlight (i.e. parallels to simplification or fallacies in clinical and medical research). In it’s current form, I feel that a number of people who have zero interest in Taylor Swift and/or American football might look at this and simply skim over it, without understanding it’s broader purpose as a commentary of sorts for medical research (e.g. ‘drawing parallels to medical research and clinical practice’).

Abstract

I feel the statement “Kelce achieved an extra mean 7.1 (95% confidence interval: -12.7, 26.9) yards per game when Swift was in attendance (n=13 games in the Swift era), compared to matched games from the pre-Swift era however this was not statistically significant (p=0.476).” in the abstract has the potential to have the reverse effect of what you are trying to achieve with this paper – in that a number of people might only take in the details of the first half of this statement and simply fall victim to the “Swift effect” (i.e. ignore the aspect that it is not a statistically significant effect). You could apply a similar notion to a follow-up statement of when Swift was not in attendance. My thoughts would be that this ‘meatier’ discussion could be incorporated in the body of the paper, but the abstract should be incredibly clear that you are saying no effect exists – given that a large proportion of your audience might just read the abstract alone.

Introduction

The point on the paper being serious and inclusion of Taylor Swift references in the paper is perhaps valuable for the review, but I would hope that it gets removed if the paper is accepted and the in-text notations (i.e. red text) be removed. I think it’s far more fun to have this in a subtle manner.

I think the introduction could benefit from highlighting some actual examples of media miscommunicating or jumping to conclusions around healthcare that isn’t supported by science. You’ve cited some references (i.e. 11-13) here but I think it would be worth highlighting these specifically to demonstrate the importance.

Methods

I think early on in the methods a more detailed (e.g. 3-4 sentence) general description of the propensity score matching technique would be helpful.

In data collection section, it would be useful to see here the exact performance metrics and game outcomes (e.g. win/loss vs. margin) that were extracted – and also to provide a justification for their use. There are a huge amount of data points for receiving players, so justifying which ones you selected would be a good addition. Reading ahead this is done later in ‘step 3’ of the next section – but it may be preferable to highlight and define the performance outcome up here. On a similar point, the selection of one variable (while justified) might be too simplistic. For example, Kelce may have had a bad receiving game, but had a good blocking game (another role of the tight end) – and hence looking at other metrics like the number of pressures allowed on the quarterback would add to this. Reading on towards the end of the discussion I see that this is a potentially selective choice – but I still wonder whether the isolated focus is appropriate.

The online news sources could use a reference or be explicitly mentioned so that readers could check these data.

It would be nice to understand why only the pre-game Elo was used to control for matches. While it is noted as the strongest confounder, presumably other factors were also relevant and matching across multiple variables would be possible? To a larger point that the paper focuses on, being selective in your matching of games via a simplistic process might also skew your conclusions. I see later on that a stepwise regression was used to come to this conclusion, but only the Chief’s pre-game Elo data is reported in the table. It would help to have the other information in the results to support this decision. For example, does inclusion of the other factors help explain even a little more of the variance above 6%? In addition, I’d include this regression table in the methods to support/justify the decision.

Results

“All games from the Swift era were successfully matched” – is there a quantitative way in which this is acceptable, or is it more qualitative in viewing the overlap of data? Some more detail here would be useful.

Any time where differences are reported (e.g. “Kelce achieved 7.1 yards per game more in the Swift-era than he did in matched games”) I think it would be worthwhile to report the variance around the mean difference (e.g. 95% CI’s). This would probably help highlight the point better, as the lack of statistical significance would suggest some wide confidence intervals in these data.

Discussion

Similar to a point in the introduction, I think the discussion would benefit from some examples or case studies from the medical and health research field that convey the point you are trying to make in the relevant sections. You have citations to support the ideas/points, but presenting some authentic and eye-catching examples would help the cause.

In line with the above comment, finishing a paragraph with the statement “Stakes in Super Bowl betting may not be critically important for society, but the phenomenon of making decisions, including financial investment, based on unsupported claims is not confined to the realm of sports” feels a little incomplete. It likely leads onto the next section, but at least providing the examples outside of the realm of sports here would be nice to see.

“or mistresses from illicit affairs” – perhaps clarify this as ‘potential’ or ‘hypothetical’, you don’t want to assume that players are having affairs.

Reviewer #2: I write to submit my report on the protocol titled “Fact or folklore? A Quasi-Experimental Investigation into Taylor Swift's Influence on American football performance”

The study determine whether the media narrative that Swift’s presence at NFL games had any impact on player or team performance – and draw parallels to scientific journalism and clinical research.

This is an interesting paper and I have few comments as follows

General comments.

The authors did not assess the quality of the resulting matched samples. There was no mention of what statistical method was used to assess the quality of matched samples. No results (numerical or graphical) were provided on whether the matching improved the similarity index between pre-Swift and post-swift era

Methods

5: 1 matching could increase bias (Poor matches) BUT High Power due to increase in sample size as the authors indicated. Sensitivity analyses must be conducted by varying the number of matched samples to determine the extent to which the impact estimate varies. For instance, will 1:1, 2:1, 3:1, and 8:1 give us the same effect size estimate? This will also help determine whether if effect size estimate is not driven by ratio of the matched sample. What determine a narrow range? What is the reference value of what authors defined as a narrow range. Sensitivity analysis of varying the ratio may be better. The optimal matching approach is also based on the ratio of the matched sample and different ratio could lead to different optimal width.

Matching with replacement can often decrease bias because controls that look similar to many treated individuals can be used multiple times. The implication of matching with replacement and matching without replacement must be discussed. Again, would the results be different if the authors have adopted matching with replacement.

Struggling to understand why the approach is propensity score matching. Propensity is the probability of intervention (Swift appearance during Football games) given certain observed covariates. It is not clear how the authors generated the propensity scores. The Elo scores are exiting database used for the matching but I am not clear how the author generated the propensity scores. It could be, say, nearest neighbor matching, caliper matching, or optimal matching based on the Elo scores, but my confusion stems from the classical definition of propensity scores. A detailed explanation must be provided

Results

The figures in the paper are not visible at all

6. PLOS authors have the option to publish the peer review history of their article (what does this mean?). If published, this will include your full peer review and any attached files.

Reviewer #1: **Yes: **Aaron Fox

Reviewer #2: No

---

## [Author Response · Author response to Decision Letter 1]

15 Apr 2025

Please see attached document for an easier-to-read, colored coded response.

Below, we have provided our responses. We indicate the beginning and end of each author response.

[BEGIN AUTHOR RESPONSE]

We would like to thank both reviewers for their time and efforts reviewing this paper and for their supportive comments and feedback. We have incorporated ALL recommendations and this has improved the quality of the paper. However, we remain open to further feedback and are happy to make revisions accordingly.

[END AUTHOR RESPONSE]

Reviewer #1: GENERAL COMMENTS

Thanks for the opportunity to read and review this paper. I quite enjoy the notion of using pop culture references alongside a little bit of a satirical take to highlight the importance of appropriate scientific communication and analysis of data, while also working in sports science myself – so I’d say this paper sits right in a niche ally of interest for me.

[BEGIN AUTHOR RESPONSE]

Thanks – we greatly appreciate that somebody like you has a unique niche interest aligned with our research was able to review this paper. Your time and positive comments are very appreciated.

[END AUTHOR RESPONSE]

Given the nature of this paper, I haven’t necessarily been overly critical in my review of the science, but in saying that this is a well written and conducted scientific study, albeit of a topic not overly important at face level. I think the premise of this paper is highlighted quite well in the authors approach, but there are opportunities to be more explicit (see specific comments on title and potential need for case study examples from medicine and health at different points). I’ve also left some detailed comments on how certain choices in the paper and communication of methodology is perhaps a little on the light side, and hence falls into a different but also important trap in scientific writing and research (i.e. isolated focus on individual outcomes, detail in replicating methods). Nonetheless, this is an enjoyable and easy read that uses a unique approach in highlighting a relevant problem to scientific communication to the media.

[BEGIN AUTHOR RESPONSE]

Thank you again. We have taken your comments, as well as Reviewer #2’s, seriously and have revised the paper accordingly. Together, this feedback improves not only the quality of the paper, but also the audience – this will increase the paper’s impact.

[END AUTHOR RESPONSE]

Specific comments on each section are provided below.

I recently received a review on one of my own papers where the reviewer provided an ‘about me’ section, which I found quite valuable in placing additional context around their comments, so it is something that I am trying to add to my own reviews:

I have a research background primarily in sports injury and performance biomechanics, but have a secondary research focus on using data and analytics approaches to characterise and improve sports performance. My work in this area has predominantly focused on netball, with a little in Australian football.

I am also open to being identified on papers I reviewed – so am signing off my review here.

Aaron Fox

Centre for Sport Research

Institute for Physical Activity and Nutrition

Deakin University

[BEGIN AUTHOR RESPONSE]

Aaron, thank you. We have had reviewers sign their name before, but I like the idea of providing some context. Unrelated to this paper, I think this can be valuable to provider context as a reviewer, as sometimes a paper is reviewed by people from different disciplines – and sometimes those reviews can be conflicting (thankfully, that is not true on this paper). Adding this context can be very helpful for the authors as they respond to comments. One of the two authors here (JMS) has a very similar background, albeit not in netball or AFL.

[END AUTHOR RESPONSE]

SPECIFIC COMMENTS

Title

I think the title should incorporate some element that highlights the broad notion that this paper is trying to highlight (i.e. parallels to simplification or fallacies in clinical and medical research). In it’s current form, I feel that a number of people who have zero interest in Taylor Swift and/or American football might look at this and simply skim over it, without understanding it’s broader purpose as a commentary of sorts for medical research (e.g. ‘drawing parallels to medical research and clinical practice’).

[BEGIN AUTHOR RESPONSE]

This is an EXCELLENT point and will definitely influence the impact of this paper. We have revised the title to: The Folklore of the “Swift” Effect – Lessons for Medical Research and Clinical Practice

[END AUTHOR RESPONSE]

Abstract

I feel the statement “Kelce achieved an extra mean 7.1 (95% confidence interval: -12.7, 26.9) yards per game when Swift was in attendance (n=13 games in the Swift era), compared to matched games from the pre-Swift era however this was not statistically significant (p=0.476).” in the abstract has the potential to have the reverse effect of what you are trying to achieve with this paper – in that a number of people might only take in the details of the first half of this statement and simply fall victim to the “Swift effect” (i.e. ignore the aspect that it is not a statistically significant effect). You could apply a similar notion to a follow-up statement of when Swift was not in attendance. My thoughts would be that this ‘meatier’ discussion could be incorporated in the body of the paper, but the abstract should be incredibly clear that you are saying no effect exists – given that a large proportion of your audience might just read the abstract alone.

[BEGIN AUTHOR RESPONSE]

This is an excellent point. We have updated the abstract to reflect these changes – and emphasize that results were not statistically significant. This compliments Reviewer #2’s request for sensitivity analysis – now that we have tried multiple matching algorithms and found no result, we can have greater confidence in these results.

[END AUTHOR RESPONSE]

Introduction

The point on the paper being serious and inclusion of Taylor Swift references in the paper is perhaps valuable for the review, but I would hope that it gets removed if the paper is accepted and the in-text notations (i.e. red text) be removed. I think it’s far more fun to have this in a subtle manner.

[BEGIN AUTHOR RESPONSE]

Yes, that was our intention. We wanted peer-reviewers to understand where we were coming from and take this paper seriously. We did not want peer-reviewers to see the humorous undertone and assume that the study was not scientifically rigorous, and prematurely dimiss/reject it for this reason. We are VERY appreciative that you recognize the value of this. We have now removed this statement.

[END AUTHOR RESPONSE]

I think the introduction could benefit from highlighting some actual examples of media miscommunicating or jumping to conclusions around healthcare that isn’t supported by science. You’ve cited some references (i.e. 11-13) here but I think it would be worth highlighting these specifically to demonstrate the importance.

[BEGIN AUTHOR RESPONSE]

The references provided here (two from Ben Goldacre) are excellent summaries of the general problem and have far too many detailed (and nuanced) examples to list. As such we have left those references intact, and provided three specific ridiculous , yet popular, claims that have appeared in mainstream media. As you suggest, these examples make the introduction more grounded in evidence that unfounded claims are everywhere and important to address.

[END AUTHOR RESPONSE]

Methods

I think early on in the methods a more detailed (e.g. 3-4 sentence) general description of the propensity score matching technique would be helpful.

[BEGIN AUTHOR RESPONSE]

This is a good idea. We have added a section to the beginning of the Methods section called “Study Overview.” This introduces the covariate matching early into the paper. It is not redundant to the other sections, since they provide further detail.

[END AUTHOR RESPONSE]

In data collection section, it would be useful to see here the exact performance metrics and game outcomes (e.g. win/loss vs. margin) that were extracted – and also to provide a justification for their use. There are a huge amount of data points for receiving players, so justifying which ones you selected would be a good addition. Reading ahead this is done later in ‘step 3’ of the next section – but it may be preferable to highlight and define the performance outcome up here. On a similar point, the selection of one variable (while justified) might be too simplistic. For example, Kelce may have had a bad receiving game, but had a good blocking game (another role of the tight end) – and hence looking at other metrics like the number of pressures allowed on the quarterback would add to this. Reading on towards the end of the discussion I see that this is a potentially selective choice – but I still wonder whether the isolated focus is appropriate.

[BEGIN AUTHOR RESPONSE]

This is a very fair point, and one that we did consider. As you mention, we did intentionally make this choice, but have now added more information about this.

• In data collection, we have added more details about which data were collected

• In “Step 3” we provided a greater justification for focusing on yards

• In the Limitations, we added a paragraph directly addressing this issue.

A more comprehensive analysis could have considered all of the many variables associated with performance, but we want to emphasize that our intent was more about verifying or refuting popular claims of the “Swift effect” (rooted in yards and win-loss records), rather than a more comprehensive analysis of multiple performance-related metrics.

If we had reason to believe that Swift really could affect other metrics of performance and that finding would provide some unique insight into sports performance or sports psychology a more thorough examination of multiple dependent outcomes would be more justified. However, as we write in discussion, there really is not good reason to believe that a single game attendee has such power, especially given the complexity of factors which influence each outcome.

[END AUTHOR RESPONSE]

The online news sources could use a reference or be explicitly mentioned so that readers could check these data.

[BEGIN AUTHOR RESPONSE]

Thanks for suggesting this – in looking through the discussion, we could now see that we regularly referred to media reports, but did not provide references. We have now added these in – they are some of the same ones referred to in the introduction, but valuable nonetheless. (We assume this is what you were referring to with this comment, but if we have missed the point, please let us know and we can resolve it.

[END AUTHOR RESPONSE]

It would be nice to understand why only the pre-game Elo was used to control for matches. While it is noted as the strongest confounder, presumably other factors were also relevant and matching across multiple variables would be possible? To a larger point that the paper focuses on, being selective in your matching of games via a simplistic process might also skew your conclusions. I see later on that a stepwise regression was used to come to this conclusion, but only the Chief’s pre-game Elo data is reported in the table. It would help to have the other information in the results to support this decision. For example, does inclusion of the other factors help explain even a little more of the variance above 6%? In addition, I’d include this regression table in the methods to support/justify the decision.

[BEGIN AUTHOR RESPONSE]

We agree with your assessment – it is surprising that other factors were not predictive of Kelce’s performance. Our analysis revealed that inclusion of other candidates failed to improve the explained variance. We have added that detail to our results section. Additionally, we have moved Table 1 from the Results section to the Methods (good suggestion).

We were especially surprised that the Chief’s pregame probability of winning was not also a factor that would influence his yards (e.g. regardless of how strong the Chiefs were, based on their elo, we would have thought that their probability of winning would also independently influence Kelce’s yards – when playing against a weak team, he should be able to perform better). The lack of additional predictors likely reflects the complexity of the game – the Chiefs success is not only dependent on Kelce, but multiple other players. Depending on the play style of the opposing team, the Chiefs may rely on Kelce more or less. Likewise, the opposing team’s defensive style may also have a major influence on Kelce’s success – regardless of how strong they are as an overall team. These are all difficult to quantify – especially with specific reference to Kelce. Furthermore, when the Chiefs played a weak team, which could theoretically allow Kelce to have a strong game, it is possible that Kelce would actually be given fewer opportunities to perform (e.g., relying on “good enough” performances from other players to minimize injury risk to their star player, especially when the Chiefs have a huge lead).

Thus, we would like to view elo as our good faith, yet imperfect, attempt to mitigate a potential confounder – while acknowledging that there are lots of other factors which could influence his performance, many of which could not be easily accounted for – or justified using our preliminary linear regression analysis.

We have added a section to our limitations section to address this.

[END AUTHOR RESPONSE]

Results

“All games from the Swift era were successfully matched” – is there a quantitative way in which this is acceptable, or is it more qualitative in viewing the overlap of data? Some more detail here would be useful.

[BEGIN AUTHOR RESPONSE]

This was also a question that Reviewer #2 asked – how to quantify the quality of the match. We have provided this text in our results:

“Following our a priori covariate matching procedure (5:1 non-replacement approach), the standardized mean differences for pre‑game Elo ratings were reduced from 0.76 to below 0.01, confirming that all Swift-era games were successfully matched with comparable historical controls. This quantitative evidence supports the assertion that the treated and control groups achieved acceptable covariate balance.”

Per Reviewer 2’s recommendations, we performed a sensitivity analysis, in which we compared multiple different matching ratios and matching algorithms (replacement vs. non-replacement. All produced similar matching and balance plots. As such, we have just presented the one for our 5:1 non-replacement match as an online supplement, since it is a good presentative plot.

From a qualitative perspective, Figure 1 provides this also. One can see that the Chiefs Elo for each Swift era game closely corresponds to the Chiefs Elo from the 5 Pre-Swift era games.

[END AUTHOR RESPONSE]

Any time where differences are reported (e.g. “Kelce achieved 7.1 yards per game more in the Swift-era than he did in matched games”) I think it would be worthwhile to report the variance around the mean difference (e.g. 95% CI’s). This would probably help highlight the point better, as the lack of statistical significance would suggest some wide confidence intervals in these data.

[BEGIN AUTHOR RESPONSE]

This is a good point, and aligns nicely with Reviewer #2’s comments. Following our added sensitivity analysis, the variance around the mean difference becomes even more evident (i.e., no matter how we analyze the data, it’s very clear that Kelce does not play better than expected when Swift is present). We have added confidence intervals into the text.

[END AUTHOR RESPONSE]

Discussion

Similar to a point in the introduction, I think the discussion would benefit from some examples or case studies from the medical and health research field that convey the point you are trying to make in the relevant sections. You have citations to support the ideas/points, but presenting some authentic and eye-catching examples would help the cause.

[BEGIN AUTHOR RESPONSE]

We have expanded the paragr

---

## [Decision Letter · Decision Letter 1]

22 Jul 2025

The Folklore of the “Swift” Effect – Lessons for Medical Research and Clinical Practice

PONE-D-24-37729R1

Dear Dr. Smoliga,

We’re pleased to inform you that your manuscript has been judged scientifically suitable for publication and will be formally accepted for publication once it meets all outstanding technical requirements.

Kind regards,

Erfan Babaee Tirkolaee, PhD

Academic Editor

PLOS ONE

Additional Editor Comments (optional):

Reviewers' comments:

Reviewer's Responses to Questions

**Comments to the Author**

1. If the authors have adequately addressed your comments raised in a previous round of review and you feel that this manuscript is now acceptable for publication, you may indicate that here to bypass the “Comments to the Author” section, enter your conflict of interest statement in the “Confidential to Editor” section, and submit your "Accept" recommendation.

Reviewer #1: All comments have been addressed

Reviewer #2: All comments have been addressed

Reviewer #3: All comments have been addressed

2. Is the manuscript technically sound, and do the data support the conclusions?

Reviewer #1: Yes

Reviewer #2: Yes

Reviewer #3: Yes

3. Has the statistical analysis been performed appropriately and rigorously? 

Reviewer #1: Yes

Reviewer #2: Yes

Reviewer #3: Yes

4. Have the authors made all data underlying the findings in their manuscript fully available?

Reviewer #1: Yes

Reviewer #2: Yes

Reviewer #3: Yes

5. Is the manuscript presented in an intelligible fashion and written in standard English?

Reviewer #1: Yes

Reviewer #2: Yes

Reviewer #3: Yes

6. Review Comments to the Author

Reviewer #1: (No Response)

Reviewer #2: The authors have addressed all comments to my satisfaction and also provided detailed clarification when the need arise. The manuscript has improved significantly

Reviewer #3: Please Acknowledge this limitation more directly in the Methods and not just in the Discussion.

Explain why receiving yards and win-loss were the only performance metrics.

7. PLOS authors have the option to publish the peer review history of their article (what does this mean?). If published, this will include your full peer review and any attached files.

Reviewer #1: **Yes: **Aaron Fox

Reviewer #2: No

Reviewer #3: **Yes: **Dr Aman Suresh T

---

## [Editor Report · Acceptance letter]

PONE-D-24-37729R1

PLOS ONE

Dear Dr. Smoliga,

I'm pleased to inform you that your manuscript has been deemed suitable for publication in PLOS ONE. Congratulations! Your manuscript is now being handed over to our production team.

Kind regards,

on behalf of

Dr. Erfan Babaee Tirkolaee

Academic Editor

PLOS ONE